# Hybrid-Fiber-Reinforced Concrete Used in Frozen Shaft Lining Structure in Coal Mines

**DOI:** 10.3390/ma12233988

**Published:** 2019-12-02

**Authors:** Zhishu Yao, Xiang Li, Taoli Wu, Long Yang, Xiaohu Liu

**Affiliations:** School of Civil Engineering and Architecture, Anhui University of Science and Technology, Huainan 232001, China; lixiang6897@126.com (X.L.); taoliwu816@163.com (T.W.); lxhhjjlty@163.com (X.L.)

**Keywords:** hybrid fiber, reinforced concrete, shaft lining structure, mechanical property, crack resistance

## Abstract

To address the cracking and leaking of concrete in frozen shaft linings in deep and thick topsoil layers in coal mines, hybrid-fiber-reinforced concrete (HFRC) was developed. First, the composition of the reference concrete was obtained by investigating high-strength concrete commonly used in shaft linings, and two dosages of polyvinyl alcohol fiber (PVAF) and polypropylene plastic steel fiber (PPSF) were obtained by the mixing test. Then, tests of early cracks of concrete were conducted; results showed that HFRC could almost avoid early cracks, exhibiting an advantage in early crack resistance. Thus, HFRC can play a significant role in improving the durability of frozen shaft linings in complex underground environments. Furthermore, a series of mechanical property tests were carried out. The results showed that the compressive strength of HFRC was similar to that of the reference concrete, but the tensile and flexural strength of HFRC was 42.7% and 35.1% higher than that of the reference concrete, respectively. Finally, an analog simulation model test of shaft linings was conducted. The new type of shaft lining structure containing hybrid fibers (HFs) exhibited plastic deformation characteristics under load, and the maximum hoop strain was −3562 με. It addressed the problem of high brittleness of frozen shaft lining structures of ordinary high-strength concrete and improved the toughness and crack resistance. HFRC is an ideal material for frozen shaft lining structures in deep and thick topsoil.

## 1. Introduction

When mining underground coal resources, it is necessary to construct tunnels from the ground surface for the transportation of coal, waste, personnel, materials, and equipment, and for ventilation and drainage. This tunnel is called a vertical shaft. A concrete wall is built to maintain its stability, which is termed the shaft lining structure.

The artificial freeze method is usually used to construct vertical shafts passing through deep and thick topsoil layers. Through artificial refrigeration, strata around the shaft is turned into frozen soil to form a frozen wall. With the protection of the frozen wall, the shaft can be excavated, and concrete is poured to construct the lining structure for support. When the shaft lining is constructed, artificial refrigeration is stopped, and the frozen wall thaws until it is completely melted. Subsequently, the shaft lining structure of the reinforced concrete protects the shaft from external pressure from water and soil [1]. Accordingly, the shaft lining structure constructed in a frozen environment is called the frozen shaft structure [2].

Owing to the strong external pressure from soil and water in deep and thick soil layers, frozen shaft lining structures have been designed with high-strength concrete, and the thickness is generally over 1.0 m [3,4]. However, in the past, water gushing or leakage occurs in the shaft lining structure, usually after the frozen soil has melted. After our analysis, we found the following reasons. The volume of concrete used to cast the shaft lining is large. The internal temperature of the concrete can reach 60–80 °C owing to hydration reaction [5,6,7], while the temperature of the frozen soil around the circular lining can reach −10 °C to −5 °C, and the temperature of air inside the circular lining can be 0 °C. Internal concrete shrinkage is caused by internal rapid cooling and the external constraint of frozen soil and formworks. Meanwhile, tensile stress also appears [8,9,10]. The tensile strength of early unset high-strength concrete is low. If the tensile stress exceeds the tensile strength because of an increase in temperature, internal cracks start forming. After the frozen soil melts, water and soil exert a high pressure on the shaft lining. This leads to the development of internal cracks that connect until the two surfaces transfix and water penetrates or floods the shaft [11,12,13]. This is a serious threat to mine safety and it is at odds with the idea of global sustainable development [14,15]. Therefore, developing a frozen shaft lining concrete with high tensile strength and excellent crack resistance is necessary [16].

The results of the present study [17,18] show that the addition of fibers can improve mechanical properties such as the tensile strength and crack resistance of concrete. It can also improve the permeability resistance. Liu et al. studied the mechanical properties of steel fiber reinforced concrete and its applications in underground structure engineering [19]. Yao et al. studied the tensile strength and crack resistance of polypropylene fiber reinforced concrete [20]. Steel fibers are easily corroded in underground water; therefore, the durability of steel fiber reinforced concrete should be studied further. Polypropylene fibers have the advantages of corrosion resistance, high tensile strength, and easy dispersion; thus, they can be a substitute for steel fibers [21,22]. It is found that the mechanical properties of multi-fiber-reinforced concrete exhibited better properties than single-fiber-reinforced concrete [23,24,25]. There is a lack of studies on the application of fiber-reinforced concrete materials in mine shaft lining structure. Therefore, an early cracking test, a series of mechanical property tests, and an analog simulation model test were conducted in this study for investigating the application and superiority of hybrid-fiber-reinforced concrete (HFRC) in frozen shaft lining structures of coal mines. The study is significant for the solution of crack and leakage problems caused by the freeze method in deep and thick topsoil layers.

## 2. Materials and Equipment

A test method for early cracking of concrete, mechanical property tests of concrete, and an analog simulation model test of shaft lining were designed to compare the material and structural mechanical properties of different concretes, so as to verify the superiority of HFRC shaft lining structures in deep and thick topsoil layers. According to the investigation of several shaft lining structures in thick and deep topsoil layers, concrete commonly used in the shaft lining had a compressive strength of over 70 MPa and excellent fluidity [26].

The Dingji coal mine, which belonged to the Huainan Mining Industry (Group) Co. Ltd., planned to construct a new auxiliary shaft to achieve safe and efficient production in the future. The designed inner and outer diameter of the lower section of the new auxiliary shaft were 8.6 m and 9.75 m, respectively (the thickness was 1.15 m). The designed compressive strength of the high-strength concrete of the shaft lining was 70 MPa. The topsoil layer around the shaft was 535 m thick. The materials of the high-strength concrete were selected as the raw materials in the following tests. The mix proportion of the high-strength concrete was selected as the reference mix proportion of concrete in this study. The size and parameters of the shaft lining were selected as the prototype size for the analog simulation model test.

### 2.1. Materials

P.O 52.5R Portland cement (Anhui Conch Group Co. Ltd. (Wuhu, China)) was selected, and its main performance indexes are listed in Table 1. Basalt crushed gravel with a continuous gradation of 5–20 mm was selected as the coarse aggregate. The crushing index of the gravel was 3.7%. Natural river sand with a fineness modulus of 2.78 was selected as the fine aggregate. The mud content of the sand was less than 1.5%. All aggregates were screened and dried. Tap water was used as mixing water and PCA–I poly carboxylic acid high performance water reducing agent with a reducing rate of 25%–30% was used as admixture. 

Fly ash was obtained from the Pingwei Thermal Power Plant (Huainan, China) and was processed into Grade I before addition. Its water demand ratio was 89%, loss on ignition was 0.95%, and fineness was 6.4%. Ultrafine slag was obtained from Sinosteel Building Materials Co., Ltd. (Beijing, China). Its powder had a specific surface area of 8000 cm^2^/g and a density of 2.89 g/cm^3^. The silicon powder had a specific surface area of 250,000–350,000 cm^2^/g. The contents of the three admixtures are listed in Table 2.

The composition of reference concrete are listed in Table 3.

Polyvinyl alcohol fibers (PVAF) and polypropylene plastic steel fibers (PPSF), obtained from Jianqing Fiber Company (Hangzhou, China), were used. Their appearances are shown in Figure 1, and their main performance indexes are listed in Table 4.

### 2.2. Equipment

A plate steel mold was used in the test for the early cracking of concrete. Its size was 600 mm × 600 mm × 63 mm. The four sides of the mold were welded from box irons and fixed with a steel bottom plate by bolts. Double row bolts with a diameter of 6 mm and spacing of 60 mm were arranged in the frame. The long and short bolt lengths were 100 mm and 50 mm, respectively. The surface of the bottom plate was laid with polyethylene film as the isolation layer.

An electric fan with a maximum wind speed of 5 m/s was used to blow the poured concrete. An anemometer with an accuracy of ±0.5 m/s was used to monitor wind speed. A thermometer with an accuracy of ±0.5 °C was used to monitor the temperature of the environment. A relative hygrometer with an accuracy of ±1% was used to monitor the relative humidity. A DJCK-2 crack width gauge with a scale of 0.01mm was used to measure the crack width.

A Quanta250 scanning electron microscope (SEM, manufactured by FEI Co. Ltd., USA) was used to observe the internal structures of both the reference concrete and the HFRC.

A 2000-kN universal testing machine (manufactured by Sanfeng instrument technology co. LTD, Changzhou, China) and Test Master software (manufactured by Lishi scientific instrument co., LTD, Shanghai, China) were used for the compressive test and the splitting tensile test.

A 1000-kN universal testing machine testing machine (manufactured by Hengruijin testing machine co. LTD, Jinan, China) and Test Master software were used for the flexural test.

A sidewall confining pressure loading device (special designed) was used for the analog simulation model test. The device had six holes for monitor wires to pass through. It was controlled by oil pressure. 32 strain gauges were prepared to paste on the surface of the model and monitor the axial strain and hoop strain during loading. The distribution of the strain gauges is shown in Figure 2.

## 3. Experiment Program

### 3.1. Mixing Test

A set of orthogonal tests were carried out to select the optimal contents of the two fibers. The collapse degree of the concrete slurry, the 28-day compressive strength and splitting tensile strength test results are listed in Table 5.

To obtain the optimum combination of the two types of fibers, the variance method was adopted to analyze the results. The results show that the slump and the compressive and tensile strength of HFRC were affected, firstly by the amount of PVAF, and secondly by the amount of PPSF. Finally, the optimum fiber combination was determined as 0.9 kg PVAF and 5.0 kg PPSF in reference concrete of one cubic meter.

### 3.2. Early Cracking Test

The tests were carried out in accordance with “Standards for Testing Methods of Long-term Performance and Durability of Ordinary Concrete (GB/T 50082-2009)” [27].

First, the environment temperature was adjusted to 20 ± 5 °C and the relative humidity to 55%–60%. Concrete slurry was then poured into the plate steel mold (Figure 3a) and was immediately spread flat. The surface was slightly higher than the mold frame. The specimen was vibrated by a plate vibrator until even, and the surface was wiped until the aggregate was not exposed.

After forming the specimen for 30 min, the wind speed of the fan was turned to 4–5 m/s at 100 mm above the center of the specimen surface, and the wind direction was adjusted to be parallel to the specimen surface and the crack inducers. Two specimens of reference concrete and HFRC were prepared for the tests. After 24 h (the timer started by adding water for stirring concrete), the cracks were measured. Crack length was measured with a steel ruler and crack width with a DJCK-2 crack width gauge (Figure 3b).

### 3.3. Mechanical Properties Test

A compressive, splitting tensile test and a flexural test were carried out in accordance with “Standard for Test Method of Mechanical Properties in Ordinary Concrete” [28]. The specimen size of the flexural test was 100 mm × 100 mm × 400 mm (Figure 4).

### 3.4. Analog Simulation Model Test

According to the analog simulation theory and the size of the test equipment, the size of the shaft lining model was determined to be 5/59 of the prototype, and raw materials, mix proportion, strength, reinforcement ratio, and other parameters were the same as the prototype.

Two 28-d reference concrete and two 28-d HFRC specimens of shaft lining were made. They were then sent to a workshop and their upper and lower annular planes were polished for smoothness. The model specimen is shown in Figure 5a. 

To obtain the stress and strain data for the model specimen during the loading process, strain gauges were placed in corresponding locations on both the internal and the external surfaces of the specimen. The inside surface of the model specimen was divided into a high and a low level, and four equidistant measurement points were marked on the inside surface at each level. Four such points were also marked on the outer surface at each level in the same way. Finally, one vertical and one horizontal gauge was placed at each measurement point to measure axial and hoop strain, respectively. Strain gauges were connected to a computer by monitoring wires. The monitoring wires attached to the strain gauges pasted on the outer surface of the model were drawn from the inside of the loading device and out through the wire holes. The load was measured using a hydraulic sensor attached to the loading device. 

Before loading onto the device, two rubber rings were placed on the upper and lower end faces of the model specimen. The two rubber rings could be deformed with the specimen in the axial direction when it was being loaded. Rigid bolts were used to constrain the plates to ensure that the model specimens remained in a plane strain state during the loading process. The confining pressure from the soil was simulated by high-pressure oil from a pump. The load was applied stably and step by step. The loading device is shown in Figure 5b.

## 4. Results and Discussion

### 4.1. Crack Resistance 

In the test method for early cracking of the reference and the HFRC, the nominal total area of cracks in the specimen were calculated using Equation (1), below [27]:(1)$$A_{cr}=\sum \omega_{i,\max}l_{i}$$
where *A_cr_* is the nominal fracture area, *ω_i,max_* is the maximum width of crack, and *i*, *l_i_* is the length of crack *i*.

The crack reduction coefficient is calculated according to Equation (2), as follows:(2)$$\eta_{cr}=\frac{A_{mcr}-A_{fcr}}{A_{mcr}}$$
where *η_cr_* is the crack reduction coefficient, *A_mcr_* is the nominal fracture area of the reference concrete, and *A_fcr_* is the nominal fracture area of the HFRC.

The calculation results of Equations (1) and (2) are listed in Table 6.

Table 6 shows that 26 cracks of different sizes appeared in the reference concrete. In contrast, only one short, narrow crack appeared in the HFRC specimen, resulting in a crack reduction coefficient of 99.74%. The HFRC showed its obvious advantages in terms of early crack resistance; almost no early cracks were formed. Therefore, HFRC was regarded as a suitable material for use in a frozen shaft lining structure constructed using the freezing method in deep thick topsoil. 

The SEM images of 28-d reference concrete and HFRC are shown in Figure 6. 

Figure 6 shows that the addition of hybrid fibers (HFs) can limit the occurrence and prevent the propagation of internal micro-cracks in concrete. The HFs effectively reduced the porosity of the concrete, avoiding the generation of connected pores so as to effectively inhibit the formation of connected cracks. This improved the crack resistance and permeability resistance of the concrete.

### 4.2. Mechanical Properties

The compressive strength and splitting tensile strength of the reference concrete and the HFRC were obtained from the mixing test. 

The flexural strength was calculated using Equation (3), below [28]. The coefficients in the formula are derived from tests.
(3)$$f_{f}=0.88Fl/bh^{2}$$
where *f_f_* is the flexural strength (MPa), *F* is the load(kN), *l* is the span between supports (mm), *b* is the thickness of the specimens (mm), and *h* is the height of the specimens (mm).

All the results of the three mechanical properties tests are listed in Table 7.

From Table 7, it can be seen that the compressive strength of HFRC was only slightly lower than that of the reference concrete; they were almost equal. This implies that HFRC can meet the requirements of early strength and design strength of frozen shaft lining, similar to commonly used concrete. The splitting tensile strength and flexural strength of the HFRC was 42.7% and 35.1%, respectively; higher than that of the reference concrete. These excellent mechanical properties are very beneficial in crack prevention and seepage resistance of the frozen shaft linings in deep and thick topsoil layers.

In addition, as shown in Figure 7, the failure form of the HFRC and the reference concrete are very different.

The reference concrete specimens fault seriously and showed obvious failure characteristics under uniaxial compression. However, the HFRC specimens were intact during fracturing, although several vertical cracks appeared, which is similar to the form of fracturing concrete under a multi-axial state. This suggests that the internal fibers of the HFRC bonded well with the concrete matrix, and the deformation of the specimens was effectively restrained. The fibers clearly improved the stress state of the concrete and prevented specimens from failure.

As shown in Figure 8, the reference concrete split into several pieces suddenly with a loud bang when the load was at maximum. However, in the HFRC specimen, only several vertical cracks appeared. It could be seen that a number of fibers connecting the two sides prevented them from cracking and deforming. 

PVAF and the PPSF played an anti-cracking role in the early and late stages of concrete failure. Mixed fibers in the concrete showed a positive hybrid effect. 

### 4.3. Bearing Capacity of Shaft Lining Model

The failure forms of shaft lining model specimens are shown in Figure 9 and Figure 10.

Figure 9 shows that the D-0 model specimen had poor ductility, one large crack, and small cracks elsewhere. There were no clear indicators of failure before the damage; it was sudden. When the shaft lining specimen broke, it made a loud noise and the concrete fell off, pulverizing the surrounding concrete and destroying the model specimen instantaneously.

Figure 10 shows that the brittleness of the D-1 model specimen was improved due to the addition of PVAF and PPSF; this greatly improved the deformation ability of the specimen. Therefore, when the model was damaged, only micro-cracks appeared in the specimen. Expansion was slow, and its effect of preventing flooding in the model was remarkable. When the model reached the ultimate bearing capacity, the specimen still had no obvious rupture surface as a whole and was still connected by fibers. Therefore, the use of HFRC as shaft lining in engineering applications could greatly improve the deformation characteristics of such a lining and the crack resistance and impermeability of its structure.

The parameters and capacities of the shaft lining model are listed in Table 8.

Table 8 shows that the ultimate capacity of the HFRC shaft lining is slightly higher than that of the reference concrete. For example, the compressive strength of the concrete of D-1 is 1% lower than that of the D-0 model specimen. Nevertheless, the capacity of D-1 is 3.8% higher than that of D-0. Evidently, the HFRC shaft lining has more advantages in bearing capacity.

After sorting through the data, the curves of the relationship between the hoop strains and confining loads of the specimens were obtained, as shown in Figure 11.

Figure 11 shows that both the curves of the D-0 inside and the D-0 outside surfaces have a linearly elastic relationship between the hoop strains and the confining loads. They have no obvious plastic deformation occurring at the destroy stage, and the value of the load is small. In contrast, although both the curves of the D-1 inside and the D-1 outside surfaces have a linear relationship during the early stage of loading, clear plastic deformation occurs as the load increases, especially before the specimen approaches the failure stage, and the maximum circumferential strain reaches −3562 με. Thus, the toughness of the shaft lining structure of the HFRC is greatly enhanced, overcoming the brittleness that is a defect of ordinary high-strength concrete. HFRC shows obvious toughness characteristics conducive to the anti-deformation requirements for use in a frozen shaft lining, and improves the reliability of the structure of the shaft lining.

The hoop stress of concrete was calculated using the damage constitutive model, according to “Code for Design of Concrete Structures” [29]. Figure 12 shows that all four curves are linearly elastic at the initial loading stage. Their slopes then tend to decrease as the load increase. D-0 curves have no obvious plastic stage in the curves during the entire loading process; however, D-1 curves show plastic properties while approaching the destroy stage. Under the same conditions, failure stress of D-1 is higher than that of D-0.

## 5. Conclusions

The early crack resistance test showed that there were 26 cracks on the reference concrete specimen but only one short and narrow crack on the HFRC specimen. The fracture reduction coefficient of the HFRC reached 99.74%, compared with the reference concrete, almost completely avoiding the generation of early cracks. It improved the durability of the frozen shaft lining in the underground complex environment in a remarkable manner.The results of the mechanical properties tests show that the HFRC and the reference concrete have similar values of compressive strength. However, the HFRC has a tensile strength and a flexural strength that were 42.7% and 35.1% higher than the reference concrete, respectively, which would be quite beneficial to crack prevention and the seepage resistance of the shaft lining in frozen, deep, and thick topsoil.When the HFRC shaft lining model was damaged, micro-cracks appeared in a limited area; macro-cracks did not appear in a large area as is common in ordinary brittle concrete. Furthermore, these micro-cracks developed slowly from the connections between fibers and the concrete matrix. Therefore, in a practical engineering application, a shaft lining structure made of HFRC can modify the brittleness of high-strength concrete and improve the crack resistance, impermeability, and durability of frozen shaft lining structures.The model test showed that the shaft lining of the HFRC had an obvious plastic flow deformation stage during the loading, with the maximum hoop strain up to −3562 με. The results indicate that the HFRC is clearly tough and can improve the structure of the shaft lining.In this study, large temperature differences between the inner and outer surfaces of the concrete were considered qualitatively, but not quantitatively. In the future, the applicability of the shaft lining structure of HFRC under the condition of large temperature differences should be studied quantitatively to improve the research of this paper.

## Figures and Tables

**Figure 1 materials-12-03988-f001:**
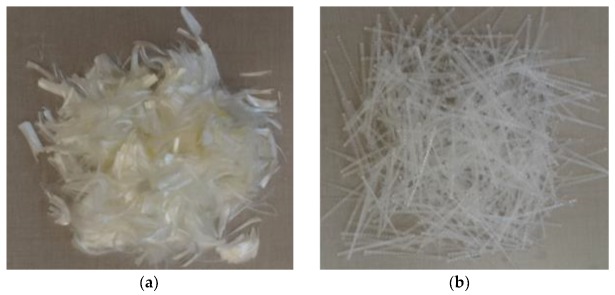
Appearance of fibers: (**a**) Polyvinyl alcohol fibers (PVAF); (**b**) Polypropylene plastic steel fibers (PPSF).

**Figure 2 materials-12-03988-f002:**
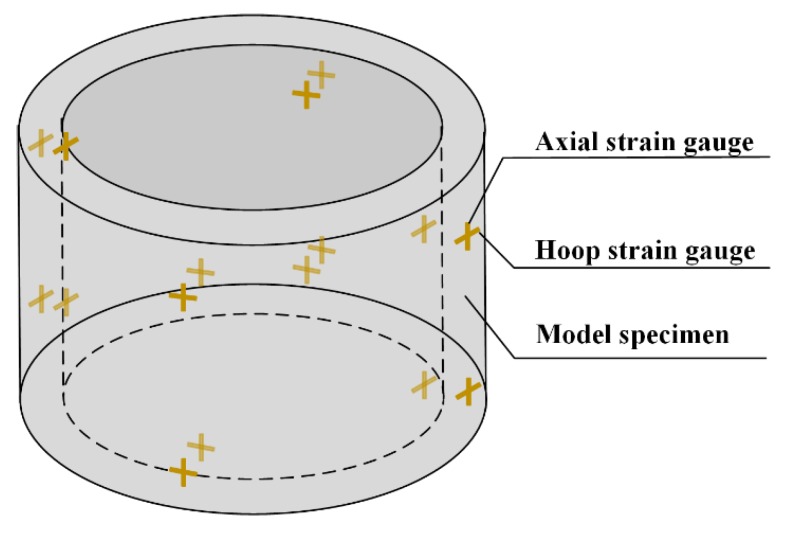
32 strain gauges pasted on the surfaces of the model specimen.

**Figure 3 materials-12-03988-f003:**
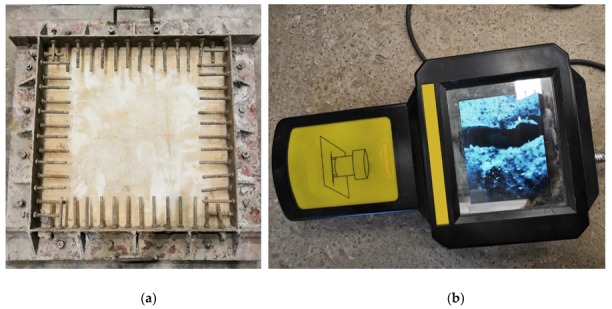
Early cracking test: (**a**) Plate steel mold; (**b**) DJCK-2 crack width gauge.

**Figure 4 materials-12-03988-f004:**
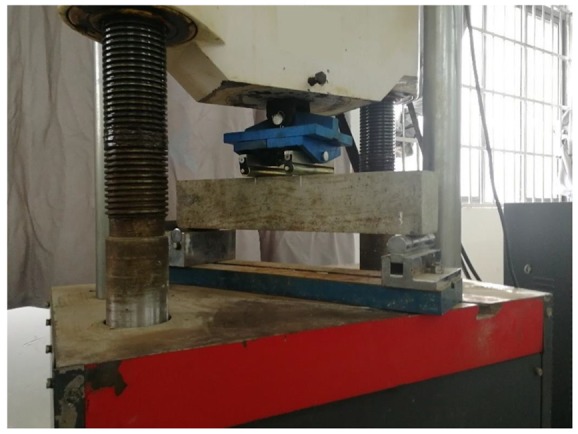
Flexural test.

**Figure 5 materials-12-03988-f005:**
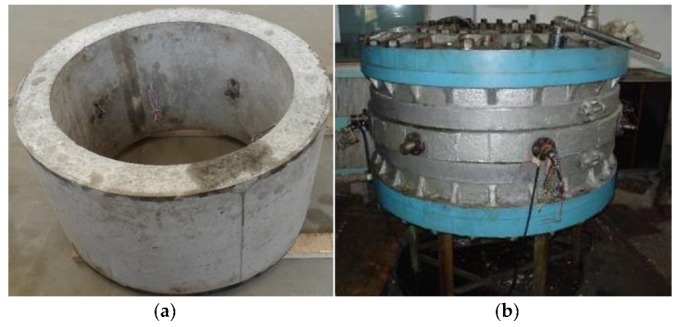
Analog simulation model test: (**a**) Model specimen; (**b**) High-pressure loading device.

**Figure 6 materials-12-03988-f006:**
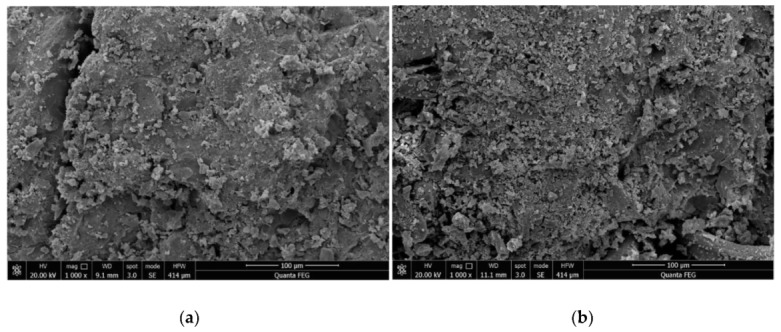
Scanning electron microscopy photos of the internal structure of: (**a**) Reference concrete and (**b**) Hybrid fiber-reinforced concrete (HFRC).

**Figure 7 materials-12-03988-f007:**
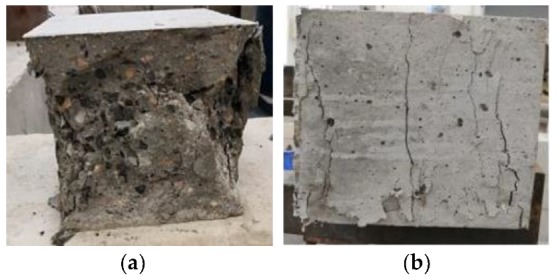
Failure form of specimens in compressive test: (**a**) Reference concrete; (**b**) HFRC.

**Figure 8 materials-12-03988-f008:**
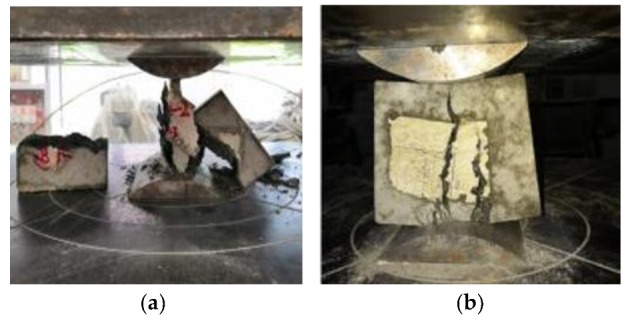
Failure form of specimens in split tensile test: (**a**) Reference concrete; (**b**) HFRC.

**Figure 9 materials-12-03988-f009:**
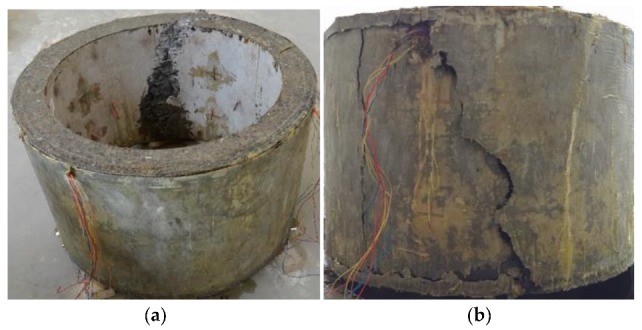
Failure characteristic of D-0 model specimen (reference concrete): (**a**) failure characteristic of inside surface (**b**) failure characteristic of outside surface.

**Figure 10 materials-12-03988-f010:**
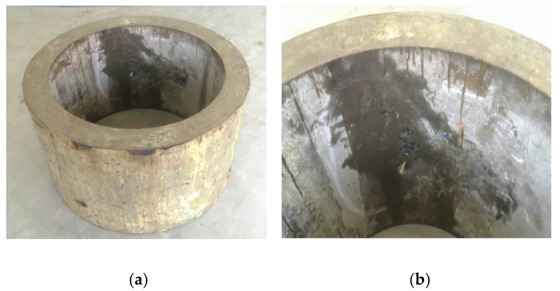
Failure characteristic of D-1 model specimen (HFRC): (**a**) an intact whole, (**b**) micro-cracks.

**Figure 11 materials-12-03988-f011:**
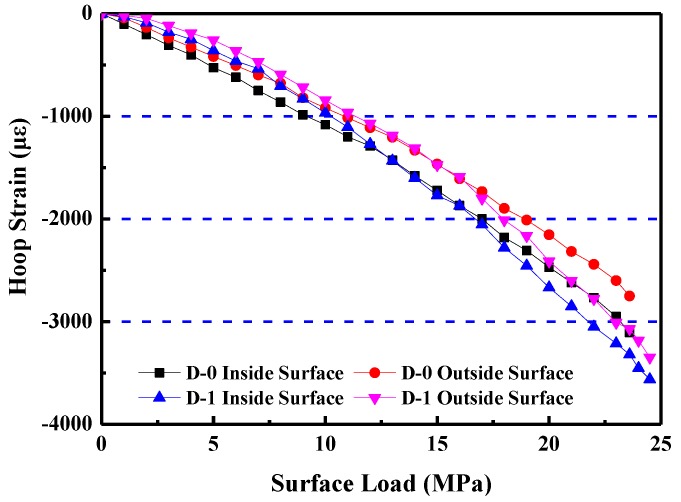
Relationship between hoop strains and surface loads of the shaft lining model specimens.

**Figure 12 materials-12-03988-f012:**
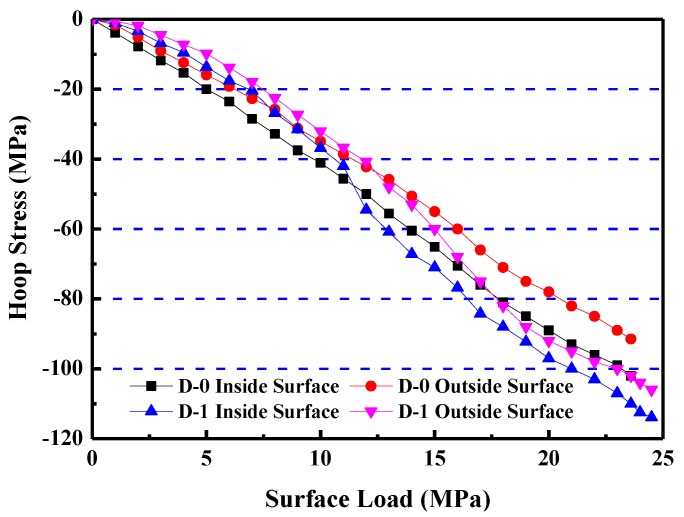
Relationship between hoop stresses and surface loads of shaft lining model specimens.

**Table 1 materials-12-03988-t001:** Performance index of P.O 52.5R cement.

Index	Stability	Setting Time		Tensile Strength/MPa		Compressive Strength/MPa	
		Initial Setting Time/min	Final Setting Time/min	3-D	28-D	3-D	28-D
P.O 52.5R	Qualified	130	265	8.1	11.9	39.5	67.8

**Table 2 materials-12-03988-t002:** Contents of admixtures (%).

Component	SiO_2_	Al_2_O_3_	Fe_2_O_3_	CaO	MgO	SO_3_
Content of fly ash	54.18	22.35	12.36	0.4	0.06	0.3
Content of ultrafine slag	32.41	9.99	1.50	40.32	6.86	2.51
Content of silicon powder	93.6	0.78	0.65	0.82	1.3	0.10

**Table 3 materials-12-03988-t003:** Composition of reference concrete (kg/m^3^).

Cement	Gravel	Sand	Water-Reducer	Water	Fly Slag	Ultrafine Slag	Silicon Powder
410	1114	626.5	11	148.5	41	78.5	20.5

**Table 4 materials-12-03988-t004:** Physical and mechanical parameters of fibers.

Fiber	Diameter/μm	Densit/g/cm^3^	Length/mm	Tensile Strength/MPa	Elongation at Break/%	Initial Modulus/GPa
PVAF	15	1.29	18	1830	6.9	40
PPSF	900	0.91	40	570	24	6.6

**Table 5 materials-12-03988-t005:** Hybrid fiber content orthogonal tests results.

NO.	PVAF Content/kg∙m^−3^	PPSF Content/kg∙m^−3^	Slump/mm	Compressive Strength/MPa	Tensile Strength/MPa
0	0	0	203	81.2	5.13
1	0.6	4.0	192	80.6	6.96
2	0.6	5.0	186	81.4	7.18
3	0.6	6.0	181	82.7	7.07
4	0.9	4.0	184	80.2	7.03
5	0.9	5.0	179	80.9	7.32
6	0.9	6.0	172	81.5	7.14
7	1.2	4.0	178	77.4	7.11
8	1.2	5.0	171	78.2	7.24
9	1.2	6.0	165	79.8	7.19

**Table 6 materials-12-03988-t006:** Results of flat slab cracking test.

Group	Max Width/mm	Max Length/mm	Average Width/mm	Number of Cracks	Total Length of Cracks/mm	*A_cr_*/mm^2^	*η_cr_*/%
Reference concrete	0.87	302	0.42	26	1475	619.4	0
Hybrid-fiber-reinforced concrete (HFRC)	0.07	23	0.07	1	23	1.61	99.74

**Table 7 materials-12-03988-t007:** Mechanical properties of concrete.

Concrete	Compressive Strength/MPa	Tensile Strength/MPa	Flexural Strength/MPa
Reference concrete	81.2	5.13	9.7
HFRC	80.9	7.32	13.1
Increase range/%	−0.37	42.7	35.1

**Table 8 materials-12-03988-t008:** Parameters and capacities of shaft lining models.

Model	Geometric Similarity Constant	Internal Diameter/mm	Outside Diameter/mm	Thickness/mm	Compressive Strength/MPa	Ultimate Capacity/MPa
D-0	5/59	729.8	925.0	97.6	81.2	23.6
D-1	5/59	729.8	925.0	97.6	80.9	24.5
Increase range/%					−0.37	3.8

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
