# Peer review of "Hybrid-Fiber-Reinforced Concrete Used in Frozen Shaft Lining Structure in Coal Mines"

_materials, 2019, doi:10.3390/ma12233988_

Round 1

Reviewer 1 Report

General comments:

The paper presents a very interesting investigation on Hybrid-fibre-reinforced Concrete Used in Frozen Shaft Lining Structure in Coal Mines.

The methodology and the presented results are clears and well developed. Only some little modifications need after the publication. The conclusions are clears and the structure of the paper is linear and easy to read.

Minor comments:

-In the Abstract:

“The results showed that.. the compressive”… The results showed that.. the new type”..please delete this repetition.

Introudction:

Please, add some lines about the actual state of the art trying to contextualize the research topic in a global sustainable scenario, thus underlying the current policies that address the research in sustainability considering thermal stress and also energy efficiency in the actual materials.   

Please, you can add these references:

-  Manni, M. Coccia, V., Nicolini, A., Marseglia, G., Petrozzi, A. “Towards Zero Energy Stadiums: the case study of the Dacia Arena in Udine, Italy”, Energies2018, 11, 2396; doi:10.3390/en11092396.

- Fabiani C Pisello A D'Alessandro A Ubertini F Cabeza L Cotana. Effect of PCM on the hydration process of cement-based mixtures: A novel thermo-mechanical investigation. Materials. 2018 23;11(6). doi: 10.3390/ma11060871.

- Marseglia, G. Rivieccio, E. Medaglia, C.M. “The dynamic role of Italian Energy strategies in the worldwide scenario”, Kybernetes 2019, Volume 48, Issue 3, pp. 636-649, https://doi.org/10.1108/K-04-2018-0199.

Page 2.. “Therefore..”. Here at the end of the introduction, you can try to add some lines to underline the aim of your paper, thus the kind of methodology used (experimental or numerical).

-Formula 1,2,3 : please specify in the text all the definition for all the letter used in the formula 1,2,3.

Reviewer 2 Report

The manuscript entitled "Hybrid-fibre-reinforced Concrete Used in Frozen Shaft Lining Structure in Coal Mines” developed hybrid-fibre-reinforced concrete to address the cracking and leaking of concrete of frozen shaft linings in deep and thick topsoil layers in coal mines. Furthermore, tests of early cracks of concrete were conducted. Two dosages of polyvinyl alcohol fibre and polypropylene plastic steel fibre were obtained by the mixing test.

The manuscript contains a clear and valuable contribution to the state of knowledge. However, the manuscript could benefit greatly from professional editing to improve technical writing and English. After a careful review, the manuscript is good in quality and therefore this reviewer recommends accepting it after considering the following comments.

Technical comments:

Abstract: “the tensile strength and flexural of HFRC” should be “the tensile and flexural strength of HFRC”. Also, “respectively” should be added at the end of this sentence. Abstract: What do you mean by “cyclic strain”. It was not mentioned in the manuscript anywhere. Introduction: The authors should increase their discussion on previous related research and highlight how their study is providing a different approach or adding significantly to what has been done. Introduction: It should be "from the ground surface to the underground". Introduction: The authors mentioned, “designed with high-strength concrete, generally over 1.0 m”. What do you mean by this number "1.0 m"? Introduction: The authors mentioned, “The internal temperature of the shaft can reach 60-80”. I think you mean "the internal temperature of concrete". Introduction: The authors mentioned “The results of the present study [14,15]”. What is the meaning of these two references here? Do you mean that you publish these results of the current study in other papers? Materials and Equipment: The authors mentioned: “concrete commonly used in the shaft lining had a compressive strength of over 70MPa and excellent fluidity”. The authors should cite a reference for this conclusion. Materials and Equipment: The authors mentioned: “The materials of the high-strength concrete was”. The sent"materials" should be "material" or "was" should be "were". Equipment: The first paragraph should be provided with a picture or schematic illustration to make it more clear and to show up the distribution of the 32 strain gauges. Figure 2(a) should be more clear to be readable. The caption of Figure 4 was repeated. The caption can be "(a) model specimen, (b) high-pressure loading device". In section 3.4: The authors mentioned “one vertical and one horizontal gauge was placed at each measurement point”. The author should explain the main purpose of two strain gages at each measuring point. For example, you can say "to measure the longitudinal and transverse strains". The authors should cite references for formula (1) and (2). Also, each term in these formulas should be defined. In section 4.2: The last sentence is unclear. Figure 11: How did the authors calculate the circumferential stress? In the elastic range, the stress can be calculated from Hook's law. What about the non-linear range of strain? Also, what is the main purpose to present strains and stresses in Figures 10 and 11? Conclusions should be written in a general format to be useful for this kind of strengthening. The authors are repeating the results with numbers. Last comment, the manuscript should be provided with line numbers to be more easy in review.
